# An Evaluation of Impostor Phenomenon in Data Science Students

**DOI:** 10.3390/ijerph20054115

**Published:** 2023-02-25

**Authors:** Lindsay Duncan, Gita Taasoobshirazi, Ashana Vaudreuil, Jitendra Sai Kota, Sweta Sneha

**Affiliations:** 1School of Data Science and Analytics, Kennesaw State University, Kennesaw, GA 30144, USA; 2Healthcare Management and Informatics, Kennesaw State University, Kennesaw, GA 30144, USA

**Keywords:** impostor phenomenon, impostor syndrome, data science, gender, motivation, perfectionism

## Abstract

Impostor Phenomenon (IP), also called impostor syndrome, involves feelings of perceived fraudulence, self-doubt, and personal incompetence that persist despite one’s education, experience, and accomplishments. This study is the first to evaluate the presence of IP among data science students and to evaluate several variables linked to IP simultaneously in a single study evaluating data science. In addition, it is the first study to evaluate the extent to which gender identification is linked to IP. We examined: (1) the degree to which IP exists in our sample; (2) how gender identification is linked to IP; (3) whether there are differences in goal orientation, domain identification, perfectionism, self-efficacy, anxiety, personal relevance, expectancy, and value for different levels of IP; and (4) the extent to which goal orientation, domain identification, perfectionism, self-efficacy, anxiety, personal relevance, expectancy, and value predict IP. We found that most students in the sample showed moderate and frequent levels of IP. Moreover, gender identification was positively related to IP for both males and females. Finally, results indicated significant differences in perfectionism, value, self-efficacy, anxiety, and avoidance goals by IP level and that perfectionism, self-efficacy, and anxiety were particularly noteworthy in predicting IP. Implications of our findings for improving IP among data science students are discussed.

## 1. Introduction

### 1.1. Impostor Phenomenon

Impostor phenomenon (IP) was first established in 1978 by Clance and Imes, who described it as a condition in which high-achieving individuals attribute their success and achievements to external factors such as excessive hard work, luck, and quota systems, and see themselves as intellectual frauds [1]. Impostor syndrome causes individuals to question and minimize themselves despite evidence of abilities, accomplishments, and skills [2,3]. As a result, those with IP tend to hold an external locus of control and external attributions [4]. Numerous studies have attempted to identify those affected by IP, the individual and contextual variables that modulate IP, and the consequences of holding IP beliefs. When it was first described, Clance and Imes believed IP was limited to women. Research, however, suggests that men also suffer from IP but that they often tend to experience it at lower levels than women [4,5]. Therefore, much of the work on IP has focused on understanding IP in women [4]. Scholars believe that damaging implicit and explicit societal messages project ideas that women are not good leaders because they are too emotional, that they are not qualified for STEM fields because they are weak at math and science, and that they are not psychologically fit for higher-level positions [6]. These factors are, in part, responsible for amplified IP among women. Unfortunately, impostor syndrome has been linked to negative psychological, behavioral, and social outcomes, including anxiety, insomnia, depression, workaholic behaviors, attrition, and withdrawal from colleagues [7,8,9,10].

IP has mostly been studied as a ‘silent career killer’ for women in business, industry, marketing, and finance [7,8]. Impostors’ exhaustive pursuits of success and paradoxical inabilities to accept success when it is attained inevitably lead to increased stress, burnout, decreased motivation, and reduced performance and fulfillment [11,12,13]. Some research has studied the impact of IP in academia and most of this research has been conducted with undergraduate students. Research shows that IP is more prevalent among undergraduate women than men [7]. Among undergraduate students, higher levels of IP were associated with higher GPAs and more time spent on academic endeavors for women but not for men [14]. A study by [15] found that undergraduates with higher IP had greater internalizations of failure. In addition, among undergraduates, perfectionism and honors program participation related to higher levels of impostor feelings [16]. Ref. [17] found a negative relationship between academic self-concept and IP among undergraduates; IP was positively related to GPA for undergraduate women, but not men. The positive link between IP and GPA for women is consistent with emerging work on IP, pointing to a positive consequence of IP being that a general fear of being exposed as a fraud can lead to additional preparation and hard work [18].

Some research has studied IP among graduate students [19,20]. Research with psychology doctoral students showed that achievement motivation was negatively correlated with IP [21]. Other research has shown that women doctoral students both suffer more from impostor syndrome and show lower research self-efficacy than men doctoral students [22]. Ref. [23] studied graduate students from a variety of disciplines learning either online or face-to-face. She found that the traditional, face-to-face graduate students had significantly higher IP scores than online graduate students; IP scores were significantly and positively correlated with anxiety for both types of learning. An additional finding was that perfectionism was the most influential predictor of IP scores for both modes of learning.

The research on IP has attempted to identify the individual difference variables that are linked to IP. For example, being Black or Hispanic [24,25,26], being a woman [1,24,27], perfectionism [16,21], motivational variables such as goal orientation [4,28], self-efficacy [21,23], and anxiety [21] have been studied as being related to IP. Most studies evaluate the impact of these variables on IP individually; a few publications have examined some of these variables together in a single study [26,29]. In the paragraphs below, we describe the research on the variables linked to IP.

Goal Orientation. Goal orientation refers to students’ reasons for engaging in various achievement behaviors in a particular situation [30]. Students with a mastery goal orientation engage in their studies to understand and learn. Their focus is on mastering the content and on personal improvement. In contrast, students with a performance goal orientation are concerned with demonstrating their ability and appearing competent [31]. Research on goal orientation further breaks down mastery and performance goals into approach and avoidance dimensions. Performance goals are divided into performance-approach (i.e., attempting to outperform others or appear competent) and performance-avoidance goals (i.e., attempting to avoid doing worse than others or appearing incompetent). Likewise, mastery goals are divided into mastery-approach (i.e., attempting to learn or improve knowledge and skills) and mastery-avoidance goals (i.e., attempting to avoid learning failures or skill decline) [32,33]. More current research on goal orientation has focused on the approach–avoidance dimensions of the two goals rather than just the mastery or performance orientations [34,35]. This research indicates that many of the negative consequences attributed to performance goals are specifically associated with performance-avoidance goals [36]. Likewise, mastery-avoidance goals are linked to negative effects, including poor motivation and performance [37]. Researchers are emphasizing the benefits of both performance-approach and mastery-approach goals and how the two goals combine to optimize motivation and performance [35]. The research on IP has found that those with greater IP are more likely to hold performance-avoidance goals [38,39]. In the present study, we evaluate if there is a relationship between IP and approach and avoidance orientations. This is the first study to examine the relationship between IP and both approach and avoidance dimensions.

Domain Identification. Domain identification is “the extent to which an individual defines the self through a role or performance in a particular domain” [40] (p. 132). Students who identify more with the domain value the domain and view it as part of their identity and self [41]. Research has shown that domain identification in STEM predicts major and career intentions [42]. The research studying the relationship between IP and domain identification is lacking. However, research on stereotype threat and domain identification suggests that students report higher domain identification when they perceive lower negative stereotype threat [43]. Accordingly, we predicted that IP would be negatively linked to domain identification.

Gender Identification. In addition to domain identification, we collected information about the extent to which a person identifies with their gender. Since IP is more prevalent among women, we hypothesized that women who more strongly identified with their gender would show higher IP and that men who more strongly identified with their gender would show lower IP.

Perfectionism. Perfectionism is defined as “a personality disposition characterized by striving for flawlessness and setting exceedingly high standards of performance accompanied by overly critical evaluations of one’s behavior” [44] (p. 171). Perfectionism has been consistently implicated across numerous studies as significantly linked to IP in that perfectionists show greater levels of IP and those with IP show greater levels of perfectionism [16,23]. Ref. [9] found that individuals with IP were inclined to criticize themselves for even the tiniest errors, severely and consistently. Ref. [2] concluded that those with IP were more concerned about mistakes, showed a higher tendency to overestimate the number of their mistakes, and were less satisfied with their results and performance.

Self-Efficacy, Anxiety, Personal Relevance, Expectancy, and Value. Many components of motivation, such as expectancy, value, anxiety, and self-efficacy, have been linked to IP. Self-efficacy refers to students’ belief that they can achieve success in a subject [45]. Research has shown that those who suffer from impostor phenomenon invariably exhibit lower levels of self-efficacy [46], which hinders productivity [47]. However, one study showed that the interaction of a moderate to frequent amount of impostor feelings with high self-efficacy increased scholarly productivity [47].

Anxiety is the debilitating tension some students experience when learning or when taking exams [48]. Feelings of impostor syndrome have been shown to be related to increased anxiety [49]; there is no research, however, determining if those with higher anxiety show greater IP.

Personal relevance is the significance of a subject to students’ goals [50]. Those who see a subject as being more personally relevant to their goals tend to be more engaged and more likely to learn [51]. No studies to date have evaluated whether personal relevance is related to IP; this study is the first to do so. Personal relevance is different from domain identification in that personal relevance is the extent to which students see the domain as being relevant to their personal goals and work, whereas domain identification is the extent to which students see the domain as part of their identity and self.

Expectancy beliefs are the degree to which an individual feels that they can be successful at a task [52]. Value beliefs are the level of importance the individual places on completing the task [52]. Both expectancy and value have been linked to course performance, career intentions, and academic aspirations [53]. The research on IP has found that those with higher IP have lower expectancies of success [54]. No research has studied the relationship between IP and value beliefs.

### 1.2. Impostor Syndrome in Data Science

We were unable to identify a research study that evaluated IP in statistics and/or data science [2,3,4]. A thorough understanding of IP in data science and statistics performance and participation is essential given that, in the last decade, there has been a huge increase in the need for individuals who can organize and analyze the vast amount of data that is being collected. This constant influx of data has put statistics, biostatistics, and data science at the forefront of coveted majors and careers. For example, Forbes Magazine listed statistics as the top 10 highest-earning college majors [55]. Glassdoor and Forbes listed Data Scientist as one of the Best Jobs in America, reporting that job openings have grown by 480% for the occupation since 2016 [56]. Fortune Magazine reported that data science is one of the fastest-growing, most critical fields of the past decade, with an expected 22% demand increase for data scientists in the next seven years [56].Thus, the need to recruit and retain diverse talent with data analytics skills has become crucial for providing varied perspectives, insights, and approaches to data analysis and statistical problem-solving needs to evaluate the constant stream of data being accumulated.

In recent years there has been a great deal of academic research and news articles that have defined IP and have reported the consequences of holding IP beliefs. If IP is prevalent among students, [57], author of Secret Thoughts of Successful Women, found that women with IP are less likely to apply for a job if they do not meet every criterion in the posting.

Further, if IP is more prevalent among women, this may be a potential reason for the gender disparity in data professional occupations. A recent report documented that less than 17% of data analytics jobs are filled by women and 12% are filled by African Americans (4%) and Hispanics (8%) [58,59]. A second and third report indicated that only about 30% of statistics-related occupations are held by women [58].

### 1.3. Present Study

This is the first study to evaluate the prevalence of IP in data science and to evaluate common variables linked to IP together in a single study evaluating data science. A major limitation of the current research on IP is that most of this research examines critical variables and their impacts on IP individually [16,29,39,45]. This study is different from the rest of the studies that are available in the literature because it evaluates multiple variables comprehensively together in one model; this allows us to determine what is of significance when multiple other variables are considered and to adjust for redundancy and/or potentially confounding variables in the model. In addition, it is the first study to evaluate the extent to which gender identification is linked to IP. Understanding the prevalence of IP and what causes IP is a necessary prerequisite for efforts to minimize its effects and increase the participation of students, particularly women and underrepresented minorities, in data science. In the present study, we examine (1) the extent to which IP exists in our sample; we evaluate the extent to which it exists for men, women, and marginalized minority groups; (2) whether gender identification is linked to IP; (3) whether there are differences in goal orientation (approach and avoid), domain identification, perfectionism, self-efficacy, anxiety, personal relevance, expectancy, and value for different levels of IP; and (4) the extent to which goal orientation (approach and avoid), domain identification, perfectionism, self-efficacy, anxiety, personal relevance, expectancy, and value are linked to IP.

## 2. Methodology

### 2.1. Participants

Eighty-six students from three different Master’s in Data Science programs from three different universities in the Southeast United States participated in the study (we emailed faculty at six universities in the Southeast United States with data science programs and received a response from faculty at three of those universities). There were approximately 120 students total enrolled in the three programs and 70% agreed to participate for a small amount of extra credit. Regarding gender, 46 identified as male and 38 identified as female; 1 was not reported; 0 responses fell outside these gender identification categories. Regarding ethnicity, 53 students reported that they were White, 20 were Asian, seven were African American, and five were Hispanic/Latino. Informed consent was collected from students, participation was voluntary, and the study was conducted in compliance with the authors’ Institutional Review Board.

### 2.2. Measures

Students were given a survey that assessed their IP, goal orientation, domain identification, perfectionism, self-efficacy, anxiety, personal relevance, expectancy, and value. The survey took students approximately 25 min to complete.

Approach and Avoidance Goal Orientations. The 12-item Achievement Goal Questionnaire (AGQ) [60] was given to students. The AGQ measures four types of goal orientation, including: mastery approach (3 items), mastery avoidance (3 items), performance approach (3 items), and performance avoidance (3 items). The performance and mastery avoidance items were combined to create a composite score, as were the performance and mastery approach items. The items were revised to focus on statistics, and students responded to the items on a five-point Likert scale, ranging from strongly disagree to strongly agree. The coefficient alpha was 0.70 for the approach items and 0.81 for the avoidance items.

Domain Identification. Five items scored on a five-point Likert scale (ranging from 1 = not at all to 5 = very much) from [61] Domain Identification Measure were revised slightly to focus on the domain of statistics and were administered to students. Internal consistency of the items for the present study was Cronbach’s alpha = 0.86.

Gender Identification. The nine gender identification items from the Social Identities and Attitudes Scale (SIAS) created by [62]. Items were on a seven-point Likert scale ranging from strongly disagree to strongly agree. The coefficient alpha for the nine items was 0.93.

Anxiety. The five anxiety items from the Statistics Motivation Questionnaire [63] were given to students to measure their anxiety for learning statistics. Students responded on a five-point Likert scale ranging from strongly disagree to strongly agree. The coefficient alpha for the five items was 0.78.

Personal Relevance. The five personal relevance items from the Statistics Motivation Questionnaire [63] were given to students to measure their belief in the relevancy of learning statistics to their personal goals. Students responded on a five-point Likert scale ranging from strongly disagree to strongly agree. The coefficient alpha for the five items was 0.77.

Perfectionism. To measure perfectionism, ref. [64] 12-item Clinical Perfectionism Questionnaire was given to students. The items measured students’ tendency towards perfectionist behaviors and students responded to the items on a four-point Likert scale ranging from not at all to all the time. Coefficient alpha for the 12 items was 0.74.

Self-Efficacy. The five self-efficacy items from the Statistics Motivation Questionnaire [63] (were given to students to measure their self-efficacy for learning statistics. Students responded on a five-point Likert scale ranging from strongly disagree to strongly agree. The coefficient alpha for the five items was 0.75.

Expectancy and Value. Three items measuring expectancy and three items measuring value [65] were administered to students. The coefficient alpha was 0.91 for the expectancy items and 0.92 for the value items.

Impostor Syndrome. The 20-item Clance Impostor Phenomenon Score (CIPS) [66] scale was used to assess IP. The CIPS is a five-point Likert scale (1 = not at all true, 5 = very true) survey that assesses IP in two ways: mean IP sum score and IP level [4,23,27]. Sum scores lower than 40 indicate few IP characteristics, 41–60 moderate, 61–80 frequent, and scores greater than 80 indicate intense levels of IP [66] (Clance, 1985). Cronbach’s alpha for the CIPS for the present study was 0.93. Consistent with previous research on IP and inventories used to measure IP, we studied IP as a categorical independent variable in our Multivariate Analysis of Variance ([23,27,67,68] and as a continuous variable in a linear regression model. We also used IP sum scores to examine the relationship between IP and gender identification.

## 3. Results

SAS 9.4 was used for all analyses. Descriptive statistics are reported in Table 1 and Table 2. To evaluate IP level differences in goal orientation (approach and avoid), perfectionism, personal relevance, anxiety, domain identification, self-efficacy, expectancy, and value, a Multivariate Analysis of Variance (MANOVA) was used. The assumptions for the MANOVA were met, including multivariate normality, lack of outliers, homogeneity of covariance matrices, and lack of collinearity. Our sample size was sufficient given the recommended sample size of 60 in G*Power (MANOVA; with effect size f^2^ = 0.15, alpha = 0.05, power = 0.80, with four groups of the independent variable and nine response variables).

To evaluate the extent to which goal orientation (approach and avoid), perfectionism, personal relevance, anxiety, domain identification, self-efficacy, expectancy, and value predicted IP, a linear regression was used. The assumptions of the linear regression (errors were normal, homoscedastic, and independent) were met. The sample size was deemed adequate (G*Power’s recommended sample size of 54 for multiple regression f^2^ = 0.35, alpha = 0.05, power = 0.80, with nine predictors).

Prevalence of IP in the Sample. Most students in the sample showed Moderate and Frequent levels of IP. A chi-square test of independence was run to determine if there was a significant association between gender and IP level; the results were not significant: chi-square (9, 86) = 11.36, *p* = 0.25. This suggested that there were not significant gender differences by IP level. When we examined counts (Table 1) almost all the men and women were at the moderate and frequent levels of IP. However, no men were at the intense level of IP, but five women held intense IP levels. When we examined IP as a continuous dependent variable using an independent samples t-test, there were no significant differences between men (M = 59.25, SD = 11.67) and women (M = 61.49, SD = 15.63) in IP sum scores, t = 0.77, *p* = 0.44.

A chi-square test of independence was run to determine if there was a significant association by race (White, Asian, Black, and Hispanic) and IP level; the results were also not significant: chi-square (9, 86) = 12.37, *p* = 0.19. This suggested that there were not significant differences in race by IP.

Multivariate Analysis of Variance. A MANOVA suggested a significant main effect for IP level: Wilks’ Lambda = 0.33, F(33, 212.83) = 2.99, *p* < 0.001, partial eta squared = 0.31 (when gender and race were included as independent variables with IP, neither was significant as a main effect or interaction). Perfectionism (F = 46.41, *p* < 0.001, partial eta squared = 0.19), value (F = 5.14, *p* = 0.003, partial eta squared = 0.16), self-efficacy (F = 4.61, *p* = 0.005, partial eta squared = 0.14), anxiety (F = 15.30, *p* < 0.001, partial eta squared = 0.36), and avoidance goals (F = 5.00, *p* = 0.003, partial eta squared = 0.16) were significant in the model.

Students with lower levels of IP (levels 1 (few) and 2 (moderate)) had significantly lower levels of perfectionism compared to those with high levels of IP (level 4 (intense)), *p* = 0.03 and *p* < 0.001, respectively. Those with higher levels of IP had significantly lower levels of value; specifically, levels 1 and 4 (*p* = 0.006), 2 and 4 (*p* = 0.02), and 3 (frequent) and 4 (*p* = 0.004) significantly differed from one another in value, with value decreasing as IP increased. For self-efficacy, those with a level 1 IP had significantly higher self-efficacy than those with a level 4 IP (*p* = 0.03). For anxiety, students with level 1 and level 2 IP had significantly higher anxiety than those with greater IP (levels 3 (1 vs. 3, *p* = 0.005; 2 vs. 3. *p* < 0.001); and 4 (1 vs. 4, *p* = 0.003, 2 vs. 4, *p* < 0.001)). For avoidance goals, those with IP levels of 2 and 3 (*p* = 0.01) significantly differed, as did those with IP levels of 2 and 4 (*p* = 0.02); specifically, those with lower levels of IP showed lower levels of avoidance goals.

Linear Regression. A linear regression with IP sum scores as the dependent variable and goal orientation (approach and avoid), domain identification, perfectionism, self-efficacy, anxiety, personal relevance, expectancy, and value as predictors was analyzed. Perfectionism (B = 0.873, *p* < 0.001), self-efficacy (B = −1.01, *p* = 0.01), and anxiety (B = −1.51, *p* < 0.001) emerged as significant in the model. The R^2^ for these three variables was evaluated at 0.49. When gender and race were included in the model, neither were significant.

Gender Identification and IP. To determine whether women who more strongly identified with their gender would show higher IP and whether men who more strongly identified with their gender would show lower IP, we used IP total sum scores and correlated them with the gender identification scores for each gender category. For men, the correlation between IP scores and gender identification was r = 0.38, *p* = 0.01. For women, the correlation between IP and gender identification was r = 0.52, *p* < 0.001. Fisher’s r to z transformation suggested that the correlations were not significantly different from one another (z = 0.80, *p* = 0.42). However, both correlations were positive and suggested a significant correlation between IP and gender identification; the correlation for women was larger (but not significantly more than for the men).

## 4. Discussion

This is the first study to evaluate the prevalence of IP in data science and to evaluate common variables linked to IP together in a single data science study. Understanding the prevalence of IP and variables involved in IP is a necessary prerequisite for efforts to minimize its effects and to increase the participation of students in data science.

We found that most of the students in our sample showed Moderate and Frequent levels of IP, indicating that IP is present among data science students. There were no significant differences in IP by gender or race. This was surprising to us, given the gender and race disparities for these groups in data science and the research on IP emphasizing the higher rates of IP among women. When we studied the many variables linked to IP using MANOVA and regression, we found that perfectionism, value, self-efficacy, anxiety, and avoidance goals played a significant role in the models. It is these variables that advisors and instructors should focus on in efforts to reduce IP among data science students. Below, we describe our findings and the practical implications.

In the MANOVA, we found that students with few and moderate levels of IP showed lower levels of perfectionism compared to those at the intense IP level. In the linear regression, perfectionism significantly and positively predicted IP. This finding is consistent with the current research on IP and perfectionism, but ours is the first to study its effects in data science and when multiple other variables are considered. Perfectionism is considered a double-edged sword in that it can motivate a person to pursue excellence, overcome adversity, and achieve success; on the other hand, perfectionism can lead an individual to fear failure, procrastinate on tasks, hold unrealistic standards, and to be excessively critical of oneself [69].

Some of the documented ways that instructors and advisors can help students overcome the negatives of perfectionism, and consequently minimize IP are: to (1) ask students to write reasonable goals for themselves at the beginning of a class or program of study; (2) allow students to understand that mistakes are part of the learning process and permit students to make mistakes that they can correct and learn from (for example, by providing formative feedback that can be used to make iterative revisions); and (3) encourage students to accept and use constructive criticism from their instructors to improve their work [70].

Values also played a role in the MANOVA; those with higher levels of IP had significantly lower levels of value. This is the first study to examine the relationship between IP and values. The students in the study were all majoring in data science, suggesting that they value the domain to some extent and see the utility and personal relevance of the domain. However, those who held higher IP levels were also less likely to see the value of their majors. Research shows that assignments that clearly articulate learning goals, highlight the real-world application of knowledge and skills, ask students to read information about the usefulness of course material, and ask students to write about how the course material relates to or is useful for one’s academic interests and career goals increases value [71]. This is something that may be of use for instructors to implement.

This study also found that students with lower IP had higher self-efficacy, which is consistent with the previous research on IP [46]. Some of the documented ways to improve student self-efficacy include: (1) model solving problems to students (and/or help students become more engaged in peer teaching) using diverse examples and presenters; (2) provide students with positive, constructive feedback on their work; (3) provide supplemental resources and activities for students for concepts and skills that have been difficult for previous cohorts to master; and (4) provide students with mastery experiences, which are opportunities to learn and practice the rules and strategies necessary to perform a task effectively [72].

We also found that students with lower IP had higher anxiety than those with higher IP. These results are opposite to what is reported in the research [49]. One possible explanation for this is that IP leads to increased preparation and an ambitious work ethic [18] which could result in decreased anxiety. Open-ended interviews in a future study could help explain these contradictory findings. Alternatively, conducting experiments on different subjects might confirm or deny the findings.

Finally, we found that those with lower levels of IP showed lower levels of avoidance of goals. Pervasive self-doubt leads to an avoidance orientation. This is a huge disadvantage for those with IP because it can lead to reduced engagement, participation, and learning [73]. Minimizing students’ fears of making mistakes and helping them to understand that mistakes are part of the learning process can help reduce an avoidance orientation [74].

It is also important to note that efforts can be made to reduce IP directly, which can help reduce perfectionism behaviors, improve value, increase self-efficacy, reduce anxiety, and decrease an avoidance goal orientation. Some of the ways that advisors and instructors can help reduce IP are to: (1) make students aware of what IP is and of its prevalence among data science students; (2) ask students to measure their own IP levels; (3) allow students to share, discuss, and document their IP experiences; and (4) educate students about the variables linked to IP in data science and how these characteristics may harm them [28].

Like all studies, the present study has limitations. Despite an a priori test of power to confirm that we met an adequate sample size for our analysis, our sample size of 86 limits the generalizability of our findings. A larger sample size would allow for stronger inferences and greater external validity. In addition, a larger sample size would allow for more sophisticated and complex modeling techniques. For example, structural equation modeling could be used to evaluate how multiple variables interact and mediate the relationship between several independent and dependent variables. A larger sample size would also allow for a more diverse sample, fulfilling the need to analyze IP in relation to the most marginalized social groups.

## 5. Conclusions

Understanding the individual difference variables linked to IP in data science and minimizing the effects of those variables can help to reduce IP among students and, consequently, increase the experiences and participation of students in data science programs. This will help ensure that we attain and maintain the talent we need for analyzing the volumes of data that inundate business, medicine, and education. We encourage future research on IP in data science using larger and more diverse samples. This would allow for greater study of IP among women and underrepresented minorities. Further, focus group interviews with students would provide a richer understanding into why IP is so prevalent among data science students. Multilevel and structural equation modeling can also be used to understand if and how IP changes over a program of study and the variables that mediate and moderate IP.

## Figures and Tables

**Table 1 ijerph-20-04115-t001:** Numbers of Students in the Different IP Categories.

IP Level	N
1—Few	5 (men = 1; women = 2)
2—Moderate	35 (men = 22; women =15)
3—Frequent	38 (men = 23; women = 16)
4—Intense	8 (women = 5; 1 not reported)

**Table 2 ijerph-20-04115-t002:** Descriptive Statistics for IP Levels for Variables in the Study.

IP Level	Mean	Std. Deviation
	Perfectionism	
1.00	30.80	6.06
2.00	31.43	4.85
3.00	34.45	4.07
4.00	38.25	4.50
Total	33.36	4.97
	Expectancy	
1.00	19.80	2.68
2.00	18.83	2.39
3.00	17.87	3.10
4.00	17.13	3.87
Total	18.30	2.92
	Value	
1.00	20.80	0.45
2.00	19.09	1.72
3.00	19.58	1.80
4.00	16.50	5.18
Total	19.16	2.39
	Gender Identification	
1.00	25.80	7.16
2.00	31.14	13.46
3.00	38.58	12.74
4.00	45.88	10.01
Total	35.49	13.47
	Domain Identification	
1.00	23.00	1.87
2.00	20.54	3.42
3.00	19.79	3.88
4.00	17.50	6.07
Total	20.07	3.96
	Self-Efficacy	
1.00	21.20	3.27
2.00	19.00	2.79
3.00	17.08	3.78
4.00	15.50	4.81
Total	17.95	3.70
	Anxiety	
1.00	18.40	2.97
2.00	17.37	3.17
3.00	12.97	3.59
4.00	11.50	2.78
Total	14.94	4.08
	Approach Goals	
1.00	23.20	1.10
2.00	27.66	6.59
3.00	27.16	7.00
4.00	29.00	5.73
Total	27.30	6.54
	Avoidance Goals	
1.00	30.40	8.26
2.00	24.11	7.99
3.00	29.50	6.85
4.00	32.75	6.11
Total	27.66	7.87
	Personal Relevance	
1.00	22.00	1.87
2.00	19.86	2.91
3.00	20.45	3.21
4.00	17.88	5.54
Total	20.06	3.37

Note: IP level of 1 = Few; 2 = Frequent; 3 = Moderate; 4 = Intense.

## Data Availability

The data for the study can be obtained by contacting G.T., Kennesaw State University, gtaasoob@kennesaw.edu.

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
