# Peer review of "An Evaluation of Impostor Phenomenon in Data Science Students"

_ijerph, 2023, doi:10.3390/ijerph20054115_

Round 1
Reviewer 1 Report
The purpose of this study was to to evaluate the presence of IP in data science and to evaluate several variables linked to IP simultaneously in a single study evaluating data science. I believe this is a meaningful study, but the following issues need attention before publication.
1. Specify what makes this article different from the rest of studies that are available in the literature.
2. Identify the gap in exiting literature, by arguing what is missing or inadequate in existing solutions and thus your study is necessary. This needs to be briefly noted in Introduction, and then further elaborated in the Literature Review, with in-depth analysis and substantiation of citations.
3. The practical implications of the study need to be further clarified. The authors studied the relationship between Imposter phenomenon and very many variables that seem to be just piling up.
Author Response
Reviewer 1
Thank you for taking the time to review our manuscript and for your helpful feedback. Below, we explain how we addressed each of your suggestions in our revised manuscript.
- Specify what makes this article different from the rest of studies that are available in the literature.
Thank you for this suggestion. In the literature review we added to the discussion on the novelty of study:
This is the first study to evaluate the prevalence of IP in data science and to evaluate common variables linked to IP together in a single study evaluating data science. A major limitation of the current research on IP is that most of this research examines critical variables and their impact on IP individually [16, 29, 39, 45]. This study is different from the rest of the studies that are available in the literature because it evaluates multiple variables comprehensively together in one model; this allows us to determine what is of significance when multiple other variables are considered and to adjust for redundancy and/or potentially confounding variables in the model. In addition, it is the first study to evaluate the extent to which gender identification is linked to IP. Understanding the prevalence of IP and what causes IP is a necessary prerequisite for efforts to minimize its effects and increase the participation of students, particularly women and marginalized groups, in data science. In the present study, we examine (1) the extent to which IP exists in our sample. We evaluate the extent to which it exists for men, women, and marginalized minority groups; (2) whether gender identification is linked to IP; (3) whether there are differences in goal orientation (approach and avoid), domain identification, perfectionism, self-efficacy, anxiety, personal relevance, expectancy, and value for different levels of IP; and (4) the extent to which goal orientation (approach and avoid), domain identification, perfectionism, self-efficacy, anxiety, personal relevance, expectancy, and value are linked to IP.
- Identify the gap in exiting literature, by arguing what is missing or inadequate in existing solutions and thus your study is necessary. This needs to be briefly noted in Introduction, and then further elaborated in the Literature Review, with in-depth analysis and substantiation of citations.
Thank you again. Please see the note above. This was added in detail in the literature review and noted briefly in the introduction of the paper.
- The practical implications of the study need to be further clarified. The authors studied the relationship between Imposter phenomenon and very many variables that seem to be just piling up.
In the discussion section, we explain our findings, how it relates to previous research, and then include practical implications of our findings (what we can do with the results to reduce IP).
Thank you!
Reviewer 2 Report
This is an interesting study focused on understanding the imposter syndrome in data science students.
1. Consider "We found that most students in the sample..."; was the sample composed of students? This information should be clear in the abstract. Perhaps it could be presented in the title.
2. Provide recent reviews in the related areas (e.g., research on IP, research on human aspects in data science, etc) to substantiate this statement "We were unable to identify a research study that evaluated IP in statistics and/or data science.".
3. Content in the methodology section could be organized in subsections. Same for results section.
4. Explicitly define the selection criteria for participants/students.
5. Limitations of the study should be acknowledged.
Specific point:
2. Method -> 2. Methodology
It is these variables - ?
Author Response
Reviewer 2
Thank you for taking the time to review our manuscript and for your helpful feedback. Below, we explain how we addressed each of your suggestions in our revised manuscript.
- Consider "We found that most students in the sample..."; was the sample composed of students? This information should be clear in the abstract. Perhaps it could be presented in the title.
We thank you for this important point! We updated the title and abstract to reflect that IP was studied among data science students.
- Provide recent reviews in the related areas (e.g., research on IP, research on human aspects in data science, etc) to substantiate this statement "We were unable to identify a research study that evaluated IP in statistics and/or data science.".
Thank you. We added three citations from reviews to substantiate this statement.
- Content in the methodology section could be organized in subsections. Same for results section.
This is very helpful! We updated the methodology section so that it is organized in subsections. We removed an unnecessary heading from the Results section so that subsections would not be needed.
- Explicitly define the selection criteria for participants/students.
We included the following statement: We emailed faculty at six universities in the Southeast United States and received a response from faculty at three of those universities.
- Limitations of the study should be acknowledged.
In the discussion section of the paper, we added a section that describes the limitations of the study.
Specific point:
- Method -> 2. Methodology
It is these variables - ?
Done!
Thank you!
Reviewer 3 Report
Some suggestions to improve the paper:
1. In the initial part it would be important to deepen the IP construct by making explicit its connections with the locus of control construct (Rotter, 1966; Lefcourt, 1991; ...) and attributive styles constructs.
2. It would be useful to explain more why it is important to study the impact of IP in the data science sector. And why it is important to analyze it in relation to the most marginalized social groups.
3. It is necessary to pay attention to the presentation of the study and recognize the limits of its external validity: it is a specific context (86 cases, 70% of students from three different Master's in Data Science programs from three different universities in the Southeast United States), which does not allow a generalization of data to all university contexts dedicated to the Data Science field
Author Response
Reviewer 3
Thank you for taking the time to review our manuscript and for your helpful feedback. Below, we explain how we addressed each of your suggestions in our revised manuscript.
- In the initial part it would be important to deepen the IP construct by making explicit its connections with the locus of control construct (Rotter, 1966; Lefcourt, 1991; ...) and attributive styles constructs.
We appreciate this! We include in the beginning of the paper that those with greater IP to hold an external locus of control and external attributions.
- It would be useful to explain more why it is important to study the impact of IP in the data science sector. And why it is important to analyze it in relation to the most marginalized social groups.
We thank you for this comment. We explain greater detail (in the literature review and present study sections) why it is important to study the impact of IP in data science. In addition, in the present study and limitations sections of the paper, we explain why it is important to evaluate IP in relation to the most marginalized groups.
- It is necessary to pay attention to the presentation of the study and recognize the limits of its external validity: it is a specific context (86 cases, 70% of students from three different Master's in Data Science programs from three different universities in the Southeast United States), which does not allow a generalization of data to all university contexts dedicated to the Data Science field.
Thank you for this suggestion. We have added a limitations paragraph to the paper in which we discuss the consequences of our small sample size on the generalizability of the results.